# Causality Preserving Chaotic Transformation and Classification using Neurochaos Learning

**Harikrishnan N. B.**
Department of Computer Science & Information Systems and APPCAIR
BITS Pilani K. K. Birla Goa Campus, India
harikrishnannb@goa.bits-pilani.ac.in

**Aditi Kathpalia**
Department of Complex Systems
Institute of Computer Science of the Czech Academy of Sciences
Prague, Czech Republic
kathpalia@cs.cas.cz

**Nithin Nagaraj**
Consciousness Studies Programme
National Institute of Advanced Studies
Bengaluru, 560012, Karnataka, India
nithin@nias.res.in

## Abstract

Discovering cause and effect variables from observational data is an important but challenging problem in science and engineering. In this work, a recently proposed brain inspired learning algorithm namely-*Neurochaos Learning* (NL) is used for the classification of cause and effect time series generated using coupled autoregressive processes, coupled 1D chaotic skew tent maps, coupled 1D chaotic logistic maps and a real-world prey-predator system. In the case of coupled skew tent maps, the proposed method consistently outperforms a five layer Deep Neural Network (DNN) and Long Short Term Memory (LSTM) architecture for unidirectional coupling coefficient values ranging from $0.1$ to $0.7$. Further, we investigate the preservation of causality in the feature extracted space of NL using Granger Causality for coupled autoregressive processes and Compression-Complexity Causality for coupled chaotic systems and real-world prey-predator dataset. Unlike DNN, LSTM and 1D Convolutional Neural Network, it is found that NL preserves the inherent causal structures present in the input timeseries data. These findings are promising for the theory and applications of causal machine learning and open up the possibility to explore the potential of NL for more sophisticated causal learning tasks.

## 1 Introduction

Despite the success of Machine Learning (ML) and Deep Learning (DL) algorithms in the field of natural language processing [1], computer vision [2], speech recognition [3], these algorithms face difficulty in interpretability and trustworthiness. One of the main reasons for this is the fact that they are merely discovering associations between 'input' and 'output' in the name of 'learning'. However, discovering associations alone is insufficient as an explanation to aid in the decision making process. Decision-making in everyday human existence relies heavily on reasoning and causal inference.

36th Conference on Neural Information Processing Systems (NeurIPS 2022).

Hence, incorporation of *causal* ability, which goes over and beyond associations (or correlations), has become an important objective in current machine learning research [4].

For time-series data, *causality* has been mathematically defined by Wiener [5]. According to Wiener, "a time series $X$ causes a time series $Y$, if the past values of $X$ contain information that help predict $Y$ above and beyond the information contained in the past values of $Y$ alone". This definition was later realized through a number of mathematical/algorithmic formulations. Granger Causality (GC) [6] was the pioneering causality estimation technique developed for time-series data. GC has been followed up with a number of extensions and inspirations, some of which are, information-theoretic causality as defined by Transfer Entropy [7] and data-compression based causality, as defined by Compression-Complexity Causality (CCC) [8]. While GC has primarily been formulated for linear (autoregressive) time-series, the latter are valid for non linear applications. These methods have been successfully employed in econometrics [9, 10], climatology [11, 12], neuroscience [13, 14] etc.

Conventional ML algorithms have not had much success in causal learning or causality based classification from time-series data [15]. They are either employed in combination with an existing time-series based causality detection technique [16, 17] or specialized methods assuming underlying causal models have been developed [18, 19, 20, 21], in order to incorporate causality learning abilities in ML.

In this work, we focus on causality detection from time-series data (without assumption of any causal model) and use a recently proposed brain inspired learning algorithm namely Neurochaos Learning [22] (NL) to learn generalized causal patterns from time series data. NL draws its inspiration from the chaotic firing of neurons in the brain [23]. NL is a rival architecture to ANNs and has shown promise in classification tasks (especially in the low training sample regime), many-a-times outperforming state-of-the-art methods. NL maps the input data into a high dimensional space which enables efficient classification. In this sense, NL shows similarity to Reservoir Computing [24] which also employs nonlinear mappings of input data. However, NL is fundamentally different from Reservoir Computing in terms of motivation, methodology, and working. NL fundamentally uses the *Topological Transitivity* property of chaos [25] and *Stochastic Resonance* [22] for learning classification tasks. Given that NL is still at a nascent stage of development, it is not well known in the AI community and most of its properties are largely unexplored. Hence, the objectives of the present study are to investigate the following:

**O1**: The efficacy of NL in *cause-effect classification* and compare the same with Deep Neural Network (DNN), 1D Convolutional Neural Network (1D CNN), and Long Short Term Memory (LSTM).

**O2**: Does success in cause-effect classification imply preservation of causality?

**O3**: Can NL use a *transfer learning* framework for cause-effect classification?

We do not build a novel causal ML algorithm, but rather explore if the existing NL architecture which does classification, has the capability to classify based on 'causal' information in the data. As a first step, we check its ability to distinguish between *cause* and *effect* time-series data. To elaborate, the problem of bivariate causality detection is considered, where the algorithm is trained to classify the 'causal' or 'driver' time-series variable as the *cause* and the time-series variable that is 'affected' or 'driven' by the former as the *effect*. We do not use any existing causality estimation method as an aid to the NL algorithm for this purpose. Further, for **O2**, we check whether features extracted from the learning architecture (in the above cause-effect classification task), preserve causality as measured by an existing time-series causality estimation method. This is important because it determines whether NL is doing a causality informed classification or not. The motivation behind many specialized causal learning algorithms that have been recently proposed is *generalized* learning, as failure in the case of distribution shifts continues to be one of the most important limitations of traditional ML algorithms [26, 27]. Hence, **O3** becomes an important objective to be looked at for an algorithm attempting to learn causal representations.

We find that a general NL architecture outperforms a five layer DNN, and LSTM architecture in cause-effect classification. Further, the features extracted using NL are found to preserve the cause-effect relationship present in input data. This, however, was not the case for DNN, 1D CNN and LSTM, probably because the classification results were not *causally* informed. The performance comparison of NL with 1D CNN and LSTM is extensively studied in the supplementary material. The findings demonstrate that even a general NL architecture is capable of some basic causal learning

and hence promising for developing more sophisticated causal ML algorithms required for different tasks.

The sections in the paper are arranged as follows: Section 2 describes the method used to do the cause-effect classification. Section 3 provides details of the simulated data used to carry out the experiments. Section 4 deals with experiments, results and discussions on simulated and real world prey-predator dataset. Section 5 addresses the limitations and scope for future work. The concluding remarks are provided in Section 6.

## 2   Neurochaos Learning

Neurochaos Learning (NL) is a novel brain inspired neuronal learning algorithm that has been recently proposed. The authors in [25, 28] state that NL is inspired from the chaotic firings of neurons in the brain and has mainly two architectures: (a) `ChaosNet` [28], (b) ChaosFEX+ML. In [29], the authors employ `ChaosNet` for continual learning. In another work [30], the authors propose deep `ChaosNet` for action recognition in videos.

Inspired by these recent developments, in this work, we employ `ChaosNet` architecture for the classification of cause-effect from observational data. The architecture consists of an input layer of Generalized Lüroth Series (GLS) neurons which are one-dimensional (1D) chaotic skew tent maps described as follows:

$$T(x) = \begin{cases} \frac{x}{b}, & 0 \le x < b, \\ \frac{(1-x)}{(1-b)}, & b \le x < 1, \end{cases} \tag{1}$$

where the skewness of the map is controlled by the parameter $b$ ($0 < b < 1$). Upon arrival of the input data/ stimulus ($x_k$, $k$-th data point in a time series), the $k$-th chaotic GLS neuron in the input layer starts firing (from the initial value $q$) until the chaotic neural trace of the neuron reaches the $\epsilon$ neighbourhood of the corresponding stimulus ($x_k$). The number of chaotic GLS neurons in the input layer is equal to the number of input stimuli (number of data points in a time series). From the neural trace thus generated from each chaotic GLS neuron, the following features are extracted:

1. *Firing time ($N$)*: The amount of time the chaotic neural trace takes to recognise the input stimulus.

2. *Firing rate ($R$)*: Fraction of time the chaotic neural trace is above the discrimination threshold $b$ so as to recognize the stimulus.

3. *Energy ($E$)*: For the chaotic neural trace $y(t)$ with firing time $N$, energy is defined as:

$$E = \sum_{t=1}^{N} |y(t)|^2. \tag{2}$$

4. *Entropy ($H$)*: For the chaotic neural trace $y(t)$, we first compute the binary symbolic sequence $Sym(t)$ as follows:

$$Sym(t_i) = \begin{cases} 0, & y(t_i) < b, \\ 1, & b \le y(t_i) < 1, \end{cases} \tag{3}$$

where $i = 1$ to $N$ (firing time). We then compute Shannon Entropy of $Sym(t)$ as follows:

$$H(Sym) = -\sum_{i=1}^{2} p_i \log_2(p_i) \text{ bits}, \tag{4}$$

where $p_1$ and $p_2$ refers to the probabilities of the symbols 0 and 1 occurring in $Sym(t)$ respectively.

For each input value $x_k$ (stimulus) of a data instance of class $c$ is mapped to a 4D vector $[N_{x_k}, R_{x_k}, E_{x_k}, H_{x_k}]$. The collection of these 4D vectors forms the ChaosFEX feature space. If the input data consists of only one stimulus (one time point from a given time series) with $Z$ classes, the mean representation vector of the $c$-th class (with $m$ data instances) is given by

$\frac{1}{m}[\sum_{j=1}^{m} N_j, \sum_{j=1}^{m} R_j, \sum_{j=1}^{m} E_j, \sum_{j=1}^{m} H_j]$. Since there is generally more than one stimulus or time point in a time series, this procedure is repeated for each time point.

In the case of `ChaosNet`, the classifier computes the cosine similarity of the ChaosFEX features extracted from the test sample with the pre-computed mean representation vectors (consisting of mean values of ChaosFEX features for each stimuli) from the training set of each class. The predicted class is assigned the label corresponding to the maximum cosine similarity. A detailed explanation of the `ChaosNet` and its working is provided in [25]. In this work, we use the `ChaosNet` architecture of NL.

## 3 Datasets

To evaluate the efficacy of `ChaosNet` and deep learning for the classification of cause-effect, we used simulated datasets from (a) Coupled autoregressive (AR) processes, (b) Coupled 1D chaotic maps in master-slave configuration (1D skew tent maps and 1D logistic maps) and real-world dataset from a (c) prey-predator system.

### 3.1 Coupled AR processes

The governing equations for the coupled AR processes are the following:
$$M(t) = a_1 M(t-1) + \gamma r(t), \tag{5}$$
$$S(t) = a_2 S(t-1) + \eta M(t-1) + \gamma r(t), \tag{6}$$
where $M(t)$ and $S(t)$ are the independent and the dependent (or the cause and effect) time series respectively at time $t$; $a_1 = 0.8$, $a_2 = 0.9$, the noise intensity $\gamma = 0.03$ and $r(t)$ is the i.i.d additive gaussian noise drawn from a standard normal distribution. The coupling coefficient $\eta$ is varied from 0 to 1 in steps of 0.1. We generate 1000 independent random trials for each value of $\eta$. Each of the data instances are of length 2000, after removing the initial 500 samples (transients) from the time series.

### 3.2 Coupled 1D Chaotic maps in Master-Slave configuration

#### 3.2.1 Coupled Skew-tent maps

The governing equations used to generate the master and slave time series for the coupled 1D skew-tent maps are the following:
$$M(t) = T_1(M(t-1)), \tag{7}$$
$$S(t) = (1-\eta)T_2(S(t-1)) + \eta M(t-1), \tag{8}$$

where $M(t)$ is the master (cause) and $S(t)$ is the slave (effect) system. $M(t)$ influences the dynamics of the slave system (equation 8). The coupling coefficient given by $\eta$ is varied from 0 to 0.9 with a step size of 0.1. $T_1(t)$, and $T_2(t)$ are skew tent maps with skewness $b_1 = 0.65$, and $b_2 = 0.47$ respectively. The initial values are chosen randomly for the master-slave system in the interval $(0, 1)$. We generate 1000 independent random trials for each value of $\eta$. Each of the data instances are of length 2000, after removing the initial 500 samples (transients) from the time series.

#### 3.2.2 Coupled Logistic maps

The 1D Logistic map is a widely used model to study population dynamics [31]. The governing dynamics for coupled logistic maps in master-slave configuration is given by:
$$M(t) = L_1(M(t-1)), \tag{9}$$
$$S(t) = (1-\eta)L_2(S(t-1)) + \eta M(t-1), \tag{10}$$
The coupling coefficient $\eta$ is varied from 0 to 0.9. $L_1(t) = A_1 \cdot L_1(t-1)(1 - L_1(t-1))$, and $L_2(t) = A_2 \cdot L_2(t-1)(1 - L_2(t-1))$, where $A_1 = 4$ and $A_2 = 3.82$. The attractor for this coupled dynamical system is provided in Figure 5b.

For both systems, 1000 data instances $(M(t), S(t))$ are generated and grouped as class-0 ($M(t)$: Cause) and class-1 ($S(t)$: Effect) respectively. Each of the data instances are of length 2000, after removing the initial 500 samples (transients) from the time series.

Table 1 gives details of the train-test split for the classification tasks for all the simulated datasets.

Table 1: Train-Test distribution for the simulated datasets.

| Class | Traindata | Testdata |
|---|---|---|
| Class-0 | 801 | 199 |
| Class-1 | 799 | 201 |
| Total | 1600 | 400 |

## 4 Experiments, Results and Discussions

In this section, we begin with a description of hyperparameter tuning for NL and DL followed by a demonstration of causality preservation by ChaosFEX for coupled AR processes, skew tent maps and logistic maps (and the failure of DL). Also, macro F1-scores for `ChaosNet` and DL for the cause-effect classification for each $\eta$ are plotted. For all results in this paper, software implementation is performed using Python 3, scikit-learn [32], keras [33], ChaosFEX toolbox [22], Multivariate Granger Causality (MVGC) toolbox [34], CCC toolbox [8] and MATLAB. Comparison of NL with 1D CNN and LSTM is extensively studied in the supplementary material.

### 4.1 Hyperparameter tuning for NL

Every ML algorithm has a set of hyperparameters that needs to be tuned for efficient performance. In the case of `ChaosNet`, there are three hyperparameters - initial neural activity ($q$), discrimination threshold ($b$), and noise intensity ($\epsilon$) [22]. The hyperparameter tuning is done only once with the traindata corresponding to $\eta = 0.4$ (Table 1) separately for the coupled AR processes and coupled skew tent maps.

For a fixed value of $b = 0.499$, and $\epsilon = 0.171$, $q$ was varied from $0.01$ to $0.98$ with a stepsize of $0.01$ for both coupled AR processes and coupled chaotic skew tent maps. In the case of coupled AR processes, a maximum average macro F1-score $= 0.605$ is obtained for $q = 0.78$. In the case of coupled skew tent maps, a maximum average macro F1-score = 1.0 was obtained for the following values of $q = [0.16, 0.26, 0.27, 0.28, 0.29, 0.30, 0.31, 0.32, 0.34, 0.36, 0.37, 0.38, 0.48, 0.51, 0.52,$ $0.56, 0.57, 0.72, 0.76, 0.77, 0.78, 0.79, 0.81, 0.82, 0.83, 0.84, 0.85, 0.86, 0.87, 0.88, 0.91, 0.92, 0.93,$ $0.94, 0.95, 0.96, 0.98]$ in a five-fold cross validation using traindata. We choose $q = 0.56$ for further experiments.

### 4.2 Deep Learning Parameters

A five layer Deep Learning architecture was used to evaluate the efficacy of cause-effect classification. The number of nodes in the input layer $= 2000$, followed by first hidden layer with $5000$ neurons and sigmoid activation function. The output from this layer is passed to second hidden layer with $500$ neurons and ReLU activation function. This is followed by $100$ neurons with ReLU activation function in the third hidden layer. The fourth hidden layer contains $30$ neurons with ReLU activation function. The output layer contains $2$ neurons with softmax activation function. Training was done for $30$ epochs.

### 4.3 Preservation of Granger Causality for coupled AR processes under a chaotic transformation

Accurate estimation of causality for coupled AR processes is ideally done by Granger Causality (GC) since GC models time series as AR processes. This is the reason GC is very popular in causal analysis of financial time series, climatology and neuroscience. We extract ChaosFEX features after a chaotic transformation of the input time series as described in Section 2. It is important to verify whether GC is preserved under such a nonlinear transformation. To test this, we perform the following experiment. For $q = 0.78$, $b = 0.499$, and $\epsilon = 0.171$, the firing time has been extracted from ChaosFEX for time series from coupled AR processes. The GC vs. coupling coefficient plot for firing time depicted in Figure 1a reveals that indeed GC is nicely preserved. The GC values shown here are obtained from 50 random trials[1]. This indicates the reliability of the chaotic transformation of NL in preserving granger

---

[1]The maximum model order setting in the MVGC toolbox was set to 30 for ChaosFEX features and to 20 for DL features.

causality and hence very desirable in applications which employ GC. Note that such a property is not available for DL (Figure 1b) making NL a very attractive candidate for causal ML applications. The experimental results pertaining to the comparative performance evaluation of NL and DL for cause-effect classification of coupled AR processes is provided in Section 5 under limitations.

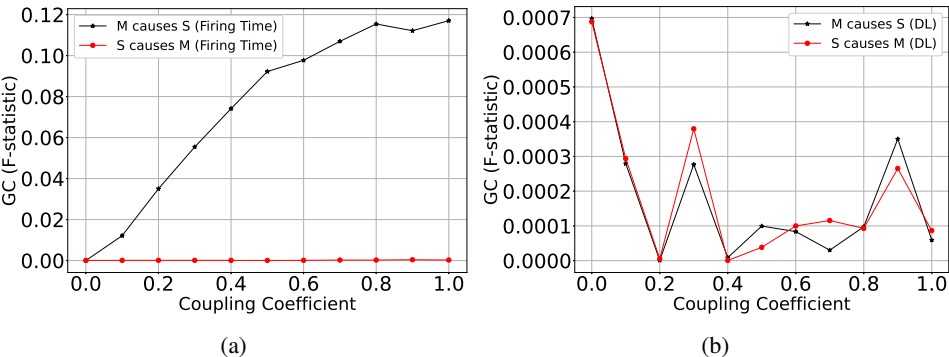

(a)          (b)

Figure 1: (a) GC vs. coupling coefficient for the firing time feature extracted from the coupled AR processes. The ChaosFEX settings are $q = 0.78$, $b = 0.499$, and $\epsilon = 0.171$. The GC F-statistic is computed from $50$ trials. (b) GC vs. coupling coefficient for DL features extracted from the fourth hidden layer of a five layer neural network. The GC F-statistic is computed from $50$ trials.

## 4.4 Classification of Cause-Effect for Coupled Skew-Tent maps in Master-Slave Configuration

In this section, we compare the efficacy of NL - `ChaosNet` architecture with a five layer DNN architecture in cause-effect classification (objective **O1**). A binary classification problem is formulated, to classify whether a given time-series is a cause or an effect. The performance of `ChaosNet` and five layer DNN (DL) for varying coupling coefficient ($\eta$) is depicted in Figure 2a.

`ChaosNet` and DL give identical performance (a macro F1-score $= 1.0$) for $\eta$ values up to $0.5$. However, for $\eta = [0.6, 0.7]$, `ChaosNet` outperforms DL. Beyond $\eta > 0.6$, the synchronization error $< 0.013$ indicating that the two time series are practically identical. Hence, classification fails as there is essentially nothing to distinguish between the two time series owing to synchronization.

## 4.5 Preservation of causality in ChaosFEX feature space

To check if causality is preserved in the ChaosFEX feature space of unidirectionally coupled skew-tent maps, we use the measure Compression-Complexity Causality (CCC) [8]. CCC is ideal for application to non linear time series, where often GC can face issues. Figure 2b shows the CCC estimates for original (raw) time series. The estimates plotted are averaged over $50$ trials with CCC parameters[2] set to $L = 100, w = 15, \delta = 50, B = 4$. As expected, the magnitude of CCC values from the master to the slave increases with increasing coupling and begins to decrease as the time-series become synchronized and effectively no transfer of information can be detected. As discussed in [8], CCC can take negative values and its magnitude denotes the strength of causation. CCC values in the direction of causation from slave to master are much lower in magnitude and remain close to zero.

CCC for the corresponding firing time feature of ChaosFEX for these coupled maps is depicted in Figure 2c. These values are also averaged over $50$ trials and computed with CCC parameters set to $L = 120, w = 15, \delta = 60, B = 2$. Here, master to slave CCC does not perfectly preserve the increasing trend with increasing values of coupling, however decreases just as the estimates for raw data, when the processes proceed to synchronization. The slave to master estimates for the coupling range $0.1 - 0.9$ are quite low in magnitude and remain close to zero as expected. Even though the estimates for zero coupling are not very close to zero or take exactly the same value and the increasing trend for increasing coupling is not perfectly preserved, CCC estimates in the direction for which coupling exists and its opposite are well differentiated and hence it can be said that ChaosFEX features do a reasonably good job in preserving causality even for skew chaotic tent maps. Surrogate

---

[2]These were chosen using the selection criteria described in [8].

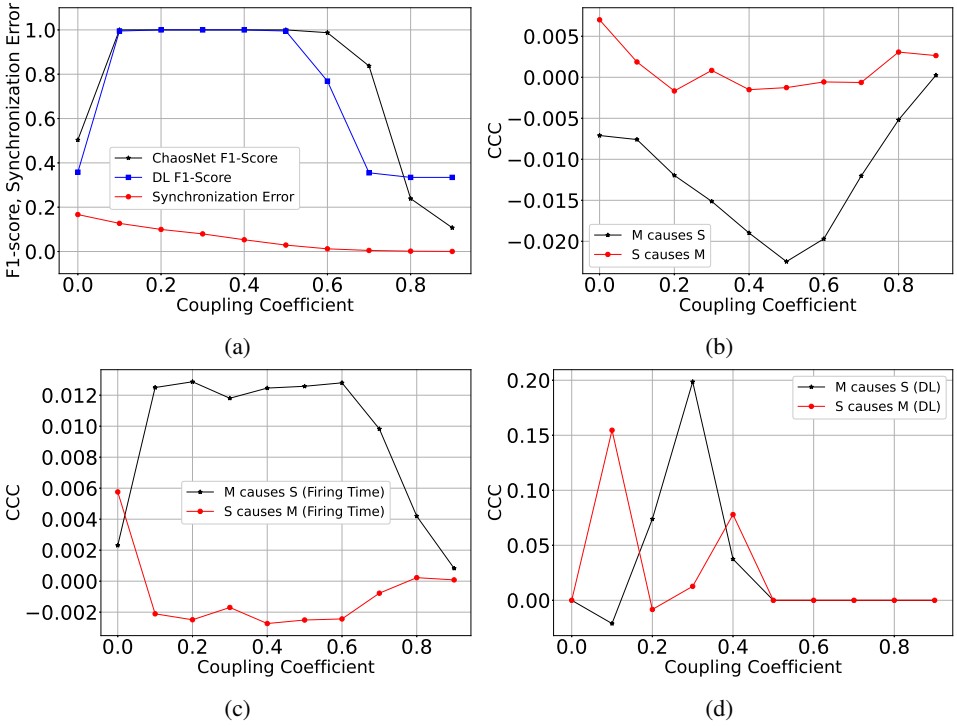

Figure 2: (a) Performance comparison of ChaosNet and five layer DNN for the classification of cause-effect for 1D coupled skew tent map in master-slave configuration. (b) CCC vs Coupling Coefficient for the raw data corresponding to 1D chaotic skew tent map in master-slave configuration. (c) CCC vs Coupling Coefficient for firing time (ChaosFEX feature) corresponding to 1D chaotic coupled skew tent maps in master-slave configuration. (d) CCC vs Coupling Coefficient for features extracted from the second last layer of five layer deep neural network corresponding to 1D chaotic coupled skew tent maps in master-slave configuration.

based causality analysis might help to reveal a more adequate picture of the differentiation and of the existence of causality, but is out of the scope of this work.

## 4.6 Transfer Learning for Cause-Effect Classification

We have demonstrated the possibility of cause-effect classification for coupled chaotic maps in master-slave configuration. However, it is interesting to explore whether we can transfer this 'learning' to scenarios where the master-slave systems are different from the ones for which the method was trained. Specifically, we shall change the skewness of both the master and slave systems from the original parameter values used in the training phase. A more drastic case of transfer learning would be to test on an entirely different nonlinear map, for example, coupled logistic maps without training afresh (using the same learned parameters as the coupled skew tent maps). These would help us determine to what extent the learning is generalizable for both NL and DL.

We consider the following cases for transfer causal learning:

- **Case I:** Train with master-slave coupled skew tent map system ($b_1 = 0.65$ , $b_2 = 0.47$) and test with master-slave coupled skew tent map system with $b_1 = 0.6$ and $b_2 = 0.4$ (classification results are in Figure 3a). The attractor for skew tent map master slave testdata with $b_1 = 0.6$ and $b_2 = 0.4$ is provided in Figure 3b.

- **Case II:** Train with master-slave coupled skew tent map system ($b_1 = 0.65$, $b_2 = 0.47$) and test with master-slave coupled skew tent map system with $b_1 = 0.1$ and $b_2 = 0.3$ (classification results are in Figure 4a). The attractor for skew tent map master slave testdata with $b_1 = 0.1$ and $b_2 = 0.3$ is provided in Figure 4b.

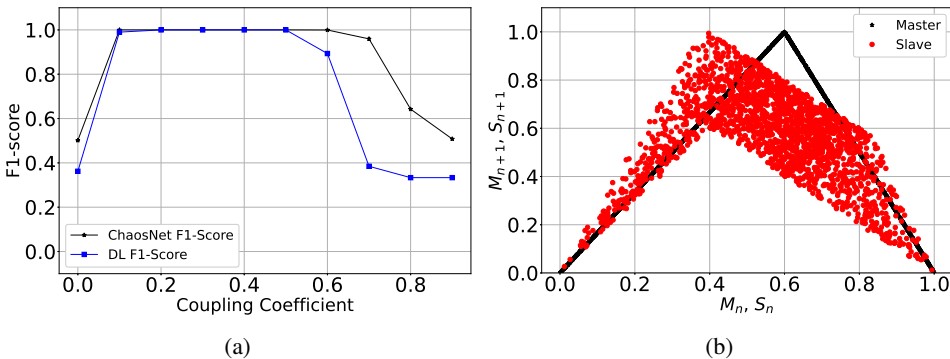

(a)                                        (b)

Figure 3: (a) Transfer learning for case I: comparative performance of ChaosNet and five layer DNN evaluated using macro F1-score for $\eta$ in the range $0$ to $0.9$ with a stepsize of $0.1$. (b) Case I: Attractor for the coupled 1D chaotic skew tent maps in master slave configuration with $b_1 = 0.6$ and $b_2 = 0.4$.

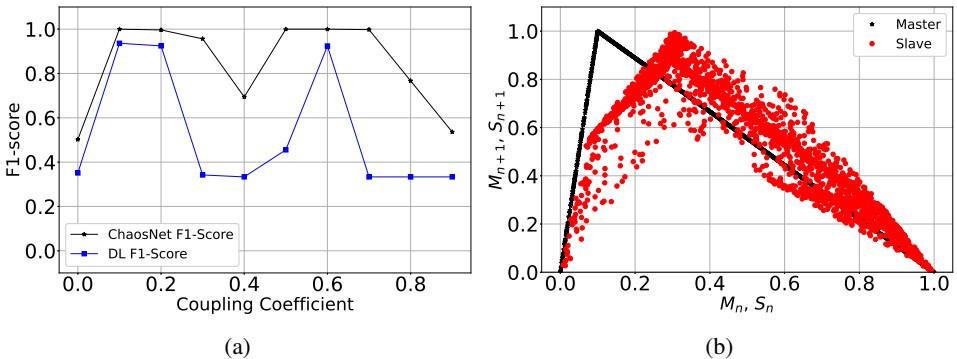

(a)                                        (b)

Figure 4: (a) Transfer Learning for Case II: comparative performance of ChaosNet and five layer DNN evaluated using macro F1-score for $\eta$ in the range $0$ to $0.9$ with a stepsize of $0.1$. (b) Case II: Attractor for the coupled 1D chaotic skew tent maps in master slave configuration with $b_1 = 0.1$ and $b_2 = 0.3$.

- **Case III:** Train with skew tent map master-slave coupled skew tent map system ($b_1 = 0.65$, $b_2 = 0.47$) and test with logistic map master-slave system with $A_1 = 4.0$ and $A_2 = 3.82$ (Figure 5a). The attractor for logistic map master slave testdata with $A_1 = 4.0$ and $A_2 = 3.82$ is provided in Figure 5b.

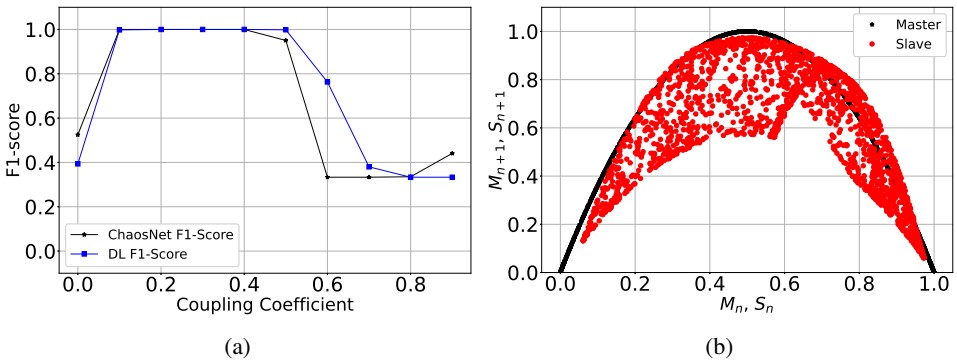

(a)                                        (b)

Figure 5: (a) Transfer Learning for Case III: comparative performance of ChaosNet and five layer DNN evaluated using macro F1-score for $\eta$ in the range $0$ to $0.9$ with a stepsize of $0.1$. (b) Case III: Attractor for the coupled 1D logistic maps in master-slave configuration with $A_1 = 4.0$ and $A_2 = 3.82$.

In the case of testing with data generated from different models, `ChaosNet` completely outperforms DL for Case I (Figure 3a) and Case II (Figure 4a) for the entire range of $\eta$. For Case III (Figure 5a), `ChaosNet` and DL shows similar trends (with DL outperforming `ChaosNet` for some values of $\eta$). A high performance of `ChaosNet` in classification shows the separability of the mean representation vectors of cause and effect.

### 4.7 Real Data

The efficacy of ChaosFEX features in cause-effect preservation was evaluated on a real world dataset from a prey-predator system as well. The data consists of 71 data points of predator (Didinium nasutum) and prey (Paramecium aurelia) populations [35, 36]. This is a system of bidirectional causation as the predator population directly influences the prey population and then itself gets influenced by a change in the prey population. It is expected that the direct causal influence from the predator to the prey should be higher than in the opposite direction.

For our analysis, initial 9 transients were removed. With the remaining 62 data points, CCC values are computed for the raw data and ChaosFEX feature (firing time). The parameters of CCC[3] used for the raw data are $L = 40, w = 15, \delta = 4, B = 8$. In the case of ChaosFEX, firing time feature was extracted for the following NL hyperparameters: $q = 0.56$, $b = 0.499$, and $\epsilon = 0.1$. The CCC parameters chosen for ChaosFEX are $L = 40, w = 15, \delta = 4, B = 4$. The results for the cause-effect preservation for the raw data and ChaosFEX firing time feature is provided in Table 2. CCC rightly captures the higher causal influence from predator to prey population and finds a lower influence in the opposite direction, for both raw data and ChaosFEX feature - firing time. As only a single data instance of predator-prey time-series is available, it is not possible to train a DL network and use its features for causality estimation in this case.

Table 2: Cause-effect preservation of the prey-predator real world data using CCC.

| Class | CCC (rawdata) | CCC (firing time) | DL |
|---|---|---|---|
| Predator $\rightarrow$ Prey | 0.1160 | 0.0484 | Unable to compute |
| Prey $\rightarrow$ Predator | -0.0210 | 0.0050 | Unable to compute |

## 5 Limitations

In the case of coupled 1D chaotic maps, NL consistently performed well up to $\eta = 0.5$ for classification. However, the same is not true for the classification of data generated from coupled AR processes. For $q = 0.78$, $b = 0.499$, and $\epsilon = 0.171$, the classification results are depicted in Figure 6. In the same figure, it can be seen that DL performance is worse than NL[4]. We have used the exact same architecture for DL as we have used for the cause-effect classification of data from coupled chaotic skew-tent maps in master-slave configure (section 4.2).

A maximum macro F1-score = 0.656 was obtained for $\eta = 1.0$ implying that `ChaosNet` was not able to find mean representation vectors that could separate the two classes. Choosing a more sophisticated classifier for NL (instead of the simplistic cosine-similarity metric) could solve this problem and improve classification results. We shall explore these possibilities in a future study.

In this research, we have shown the classification and preservation of causality for unidirectional causation of two variables. A detailed study needs to be undertaken for the classification and causal discovery of coupled high dimensional systems and real world datasets in the future.

---

[3]These were chosen using the selection criteria described in [8].

[4]We have performed some amount of hyperparmater tuning for DL architecture, however a more extensive tuning needs to be performed.

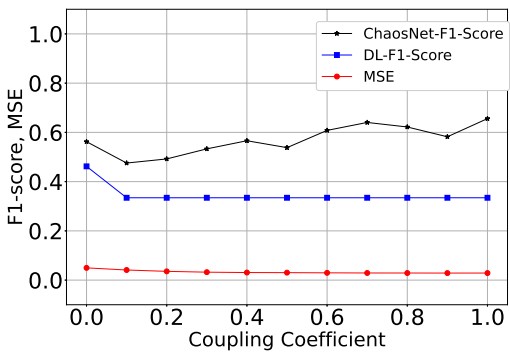

Figure 6: F1-score vs. coupling coefficient for the classification of the coupled AR processes using ChaosNet.

# 6 Conclusion

In this work, Neurochaos Learning architecture - `ChaosNet` has been used in the classification of cause and effect variables from time series generated from coupled chaotic maps and stochastic autoregressive processes. `ChaosNet` outperforms a five layer deep learning architecture and LSTM in the case of both chaotic tent maps and AR processes (objective **O1**). In the case of AR processes, 1D CNN performs better than `ChaosNet` for several values of coupling. Whereas, in the case of coupled chaotic skew-tent map, `ChaosNet` outperforms all the other methods including 1D CNN. Causality testing using Granger Causality (for coupled AR processes) and Compression-Complexity Causality (for coupled chaotic systems and for a real-world prey-predator system) on the firing times extracted from the chaotic neural traces reveals the preservation of cause-effect in the NL feature extracted space (objective **O2**). Features extracted from DNN, 1D CNN and LSTM failed to preserve the cause-effect relationship as measured by GC and CCC for coupled AR processes and skew tent map master slave system. This implies that the classification was not causally informed despite 1D CNN showing superior classification performance in some cases. Further, the efficacy of the proposed method was observed in *transfer learning* of the classification of cause-effect from the master-slave time series generated from different chaotic unimodal maps (skew-tent maps with different skews and logistic map with different parameters) (objective **O3**). This motivates future research direction of NL in lifelong learning framework, classification of cause-effect using `ChaosNet` on real world datasets and building more sophisticated causal learning algorithms using NL for specific tasks.

The preservation of causality can be attributed to the rich properties of the nonlinear chaotic transformation of GLS neurons in NL (`ChaosNet`). Unlike traditional ANNs, NL is intrinsically a nonlinear deterministic algorithm that performs a point-by-point chaotic transformation, in fact, a nonlinear embedding of the input raw features in to a high dimensional space. *Deterministic Chaos* combines the best of both the worlds - *pseudo-randomness* and *determinism*. The ergodic, 'random-like' structure of the chaotic neural traces enables an effective transformation of the input data (stimuli) preserving causality that is inherent in the input space and at the same time ensuring separability in the chaotic feature space for efficient classification. The codes used in this study are available here: `https://github.com/HarikrishnanNB/cause-effect-preservation-nl`.

## Acknowledgements

The authors gratefully acknowledge the National Institute of Advanced Studies Consciousness Studies Programme Tata Trusts Project on Kashmir Saivism, Causality & Information Theory TET/MUM/INN/NIAS/2017-2018/051-al 2018 March to 2022 March. AK is thankful to the funding provided by the Czech Science Foundation, Project No. GA19-16066S and by the Czech Academy of Sciences, Praemium Academiae awarded to M. Paluš.

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
