# Supplementary Material for 'Causality Preserving Chaotic Transformation and Classification using Neurochaos Learning'

Harikrishnan N. B.[*1], Aditi Kathpalia[†2], and Nithin Nagaraj[‡3]

[1]Department of Computer Science & Information Systems and APPCAIR, BITS Pilani K. K. Birla Goa Campus, India
[2]Department of Complex Systems, Institute of Computer Science of the Czech Academy of Sciences, Prague, Czech Republic
[3]Consciousness Studies Programme, National Institute of Advanced Studies, Indian Institute of Science Campus, Bengaluru, 560012, Karnataka, India

This is the supplementary information pertaining to the main manuscript. In this supplementary material, we provide the comparative performance of Neurochaos Learning with Deep Neural Network, 1D Convolutional Neural Network (1D CNN), and Long Short term Memory (LSTM) for evaluation of cause-effect classification of timeseries data generated from coupled chaotic master-slave system and autoregressive (AR) processes. We also check whether each of these architectures are able to preserve cause-effect relationship between the corresponding features extracted from the original cause and effect time series.

## 1 Datasets

To evaluate the efficacy of Neurochaos Learning (NL: `ChaosNet`) and deep learning algorithms for the classification of cause-effect, we used simulated datasets from (a) coupled autoregressive (AR) processes, and (b) coupled 1D chaotic skew tent-maps in master-slave configuration.

### 1.1 Coupled AR processes

The governing equations for the coupled AR processes are the following:

$$M(t) = a_1 M(t-1) + \gamma r(t), \tag{1}$$
$$S(t) = a_2 S(t-1) + \eta M(t-1) + \gamma r(t), \tag{2}$$

where $M(t)$ and $S(t)$ are the independent and the dependent (or the cause and effect) time series respectively; $a_1 = 0.8$, $a_2 = 0.9$, the noise intensity $\gamma = 0.03$ and $r(t)$ is independent and identically distributed additive Gaussian noise drawn from a standard normal distribution. The coupling coefficient $\eta$ is varied from 0 to 1 in steps of 0.1. We generated 1000 independent random trials for each value of $\eta$. Each of the data instances are of length 2000, after removing the initial 500 samples (transients) from the time series.

### 1.2 Coupled 1D Chaotic maps in Master-Slave configuration

#### 1.2.1 Coupled Skew-tent maps

The governing equations used to generate the master and slave time series for the coupled 1D skew-tent maps are the following:

$$M(t) = T_1(M(t-1)), \tag{3}$$
$$S(t) = (1-\eta)T_2(S(t-1)) + \eta M(t-1), \tag{4}$$

[*]harikrishnannb@goa.bits-pilani.ac.in
[†]kathpalia@cs.cas.cz
[‡]nithin@nias.res.in

where $M(t)$ is the master (cause) and $S(t)$ is the slave (effect) system. $M(t)$ influences the dynamics of the slave system (equation 4). The coupling coefficient given by $\eta$ is varied from 0 to 0.9 with a step size of 0.1. $T_1(t)$, and $T_2(t)$ are skew tent maps with skewness $b_1 = 0.65$, and $b_2 = 0.47$ respectively. The initial values are chosen randomly for the master-slave system in the interval $(0, 1)$. We generated 1000 independent random trials for each value of $\eta$. Each of the data instances are of length 2000, after removing the initial 500 samples (transients) from the time series.

For both systems (coupled AR processes and Coupled Skew-tent map), 1000 data instances $(M(t), S(t))$ are generated and grouped as class-0 ($M(t)$: Cause) and class-1 ($S(t)$: Effect) respectively.

Table S1 gives details of the train-test split for the classification tasks for all the simulated datasets.

Table S1: Train-Test distribution for the simulated datasets.

| Class | Traindata | Testdata |
|---|---|---|
| Class-0 | 801 | 199 |
| Class-1 | 799 | 201 |
| Total | 1600 | 400 |

# 2 Experiments and Results

In the main manuscript, we compare the performance of NL with a five layer deep neural network. In this supplementary document, we extend the comparison of NL with 1D CNN and LSTM. The architecture details of 1D CNN and LSTM used in this research are provided in Table S2 and Table S3 respectively.

Table S2: 1D CNN Architecture details.

| Layer (Type) | Output Shape | Parameters |
|---|---|---|
| Conv1D | (None, 1998, 32) | 128 |
| MaxPooling | (None, 999, 32) | 0 |
| Flatten | (None, 31968) | 0 |
| Dense | (None, 32) | 1023008 |
| Dropout | (None, 32) | 0 |
| Dense | (None, 2) | 66 |
| Total Parameters | 1,023,202 | |
| Trainable Parameters | 1,023,202 | |

Table S3: LSTM architecture details.

| Layer (Type) | Output Shape | Parameters |
|---|---|---|
| LSTM | (None, 72) | 597024 |
| Dense | (None, 2) | 146 |
| Total Parameters | 597170 | |
| Trainable Parameters | 597170 | |

## 2.1 Classification/preservation of cause-effect relationship for coupled AR processes

In this section, we compare the efficacy of NL - `ChaosNet` architecture with a five layer DNN architecture, 1D CNN and LSTM in cause-effect classification (objective **O1** - please refer to main manuscript). A binary classification problem is formulated, to classify whether a given time-series is a cause or an effect. The performance of `ChaosNet` with five layer DNN (DL), 1D CNN and LSTM for varying coupling coefficient ($\eta$) is depicted in Figure S1.

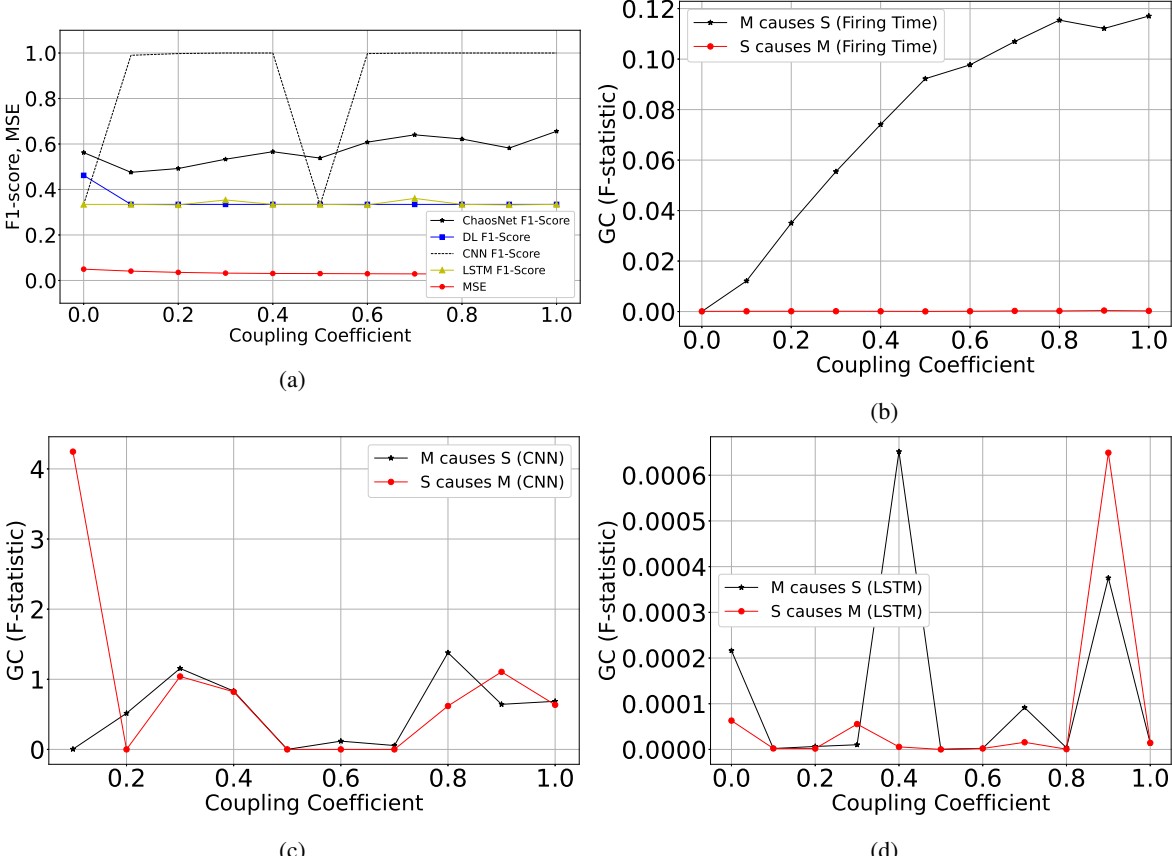

Figure S1: (a) Performance comparison of ChaosNet with five layer DNN, 1D CNN, and LSTM for the classification of cause-effect for timeseries data generated from coupled AR processes. (b) GC vs Coupling Coefficient for the firing time feature (ChaosFEX feature) extracted from the input layer of NL. (c) GC vs Coupling Coefficient for features extracted from the second last layer of 1D CNN architecture. (d) GC vs Coupling Coefficient for features extracted from the second last layer of LSTM architecture.

Considering only classification of *cause* from *effect*, 1D CNN outperforms NL, DL, and LSTM (see Fig. S1(a)). We have also examined the causality preserving property of NL, DL, CNN, and LSTM for coupled AR processes as measured by GC (Fig. S1(b)-(d)). From the above graphs, clearly *NL firing time feature preserves the cause-effect relationship consistently for varying coupling coefficients*. On the other hand, DL, 1D CNN and LSTM *completely fail* to capture the correct cause-effect relationship between the Master and Slave time series.

## 2.2 Classification/preservation of cause-effect relationship for coupled skew tent map master slave system

In this section, we compare the efficacy of NL - `ChaosNet` architecture with a five layer DNN architecture, 1D CNN and LSTM in cause-effect classification (objective **O1** - please refer to the main manuscript). A binary classification problem is formulated, to classify whether a given time-series is a cause or an effect. The performance of `ChaosNet` with five layer DNN (DL), 1D CNN and LSTM for varying coupling coefficient $(\eta)$ is depicted in Figure S2.

From the above graphs in Fig. S2(a), we infer that NL outperforms DL, CNN, and LSTM architectures. In the case of CNN, for coupling coefficient = 0, an F1-score = 1.0 is yielded which is actually incorrect. A coupling coefficient = 0 means there is no causal influence from master to slave and hence the classifier should in fact yield a performance score of 50% as the two time series are causally independent from each other (ChaosNet, DL and LSTM are on target here). This implies that CNN is not truly making causally informed classification but rather 'memorising' the data. The performance of LSTM was poor compared to all other architectures. In future, we plan to do extensive hyperparameter tuning for LSTM, which may result in improved performance.

We have also examined the causality preserving property of NL, DL, CNN, and LSTM as measured using Compression-Complexity Causality (CCC) (Fig. S2(b)-(d)). The parameters of CCC used for each case are the following:

- NL: L = 120, w = 15, $\delta = 60$, B = 2

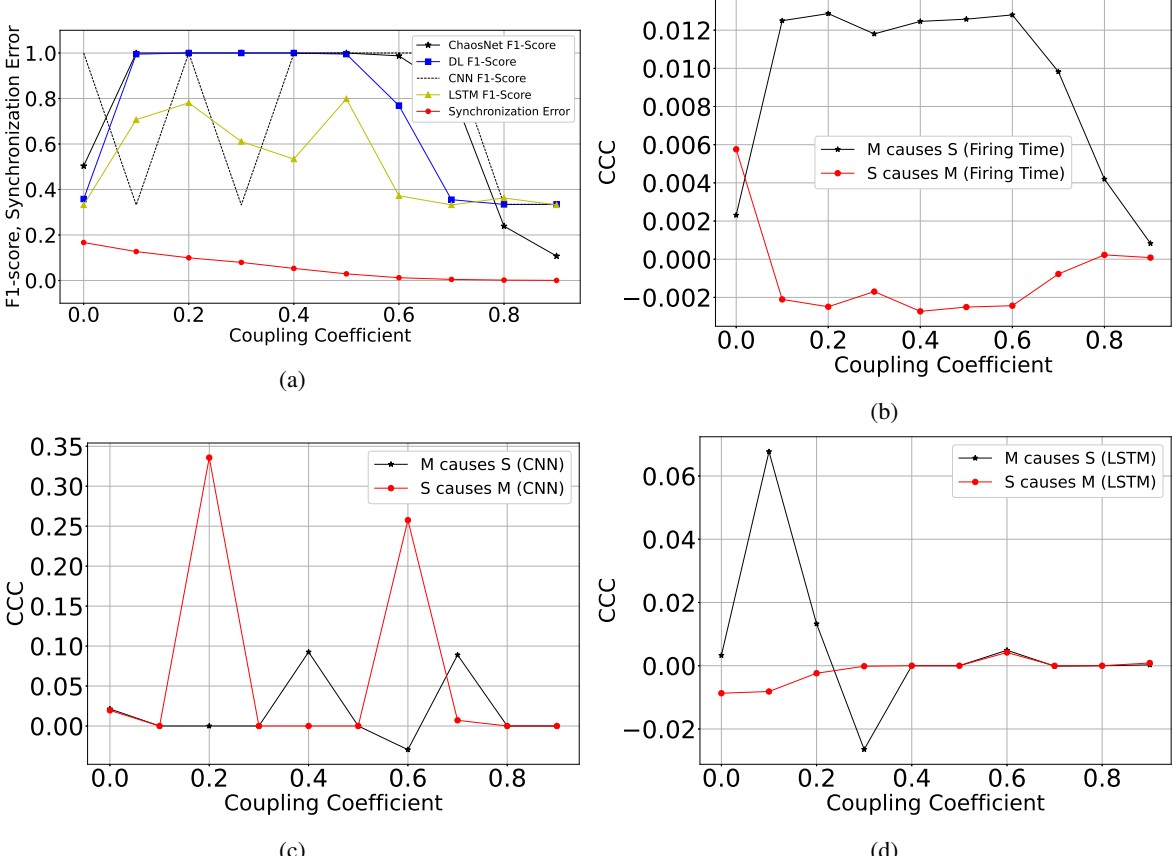

(a)

(b)

(c)

(d)

Figure S2: (a) Performance comparison of ChaosNet with five layer DNN, 1D CNN, and LSTM for the classification of cause-effect for timeseries data generated from coupled 1D skew-tent map master-slave configuration. (b) CCC vs Coupling Coefficient for the firing time feature (ChaosFEX feature) extracted from the input layer of NL. (c) CCC vs Coupling Coefficient for features extracted from the second last layer of 1D CNN architecture. (d) CCC vs Coupling Coefficient for features extracted from the second last layer of LSTM architecture.

- DL: L = 15, w = 10, $\delta = 2$, B = 2

- CNN: L = 20, w = 10, $\delta = 2$, B = 2

- LSTM: L = 25, w = 20, $\delta = 4$, B = 2

It can be seen that only NL is able to preserve the cause-effect relationship *consistently* for varying values of coupling coefficient. CNN and LSTM preserve causal relationships only for some values of coupling coefficient, but not consistently across the entire range of coupling coefficients. Furthermore, it was found that different random trials of CNN and LSTM led to different results - thereby indicating the inconsistency and unreliability of these methods for causality preservation. Only NL is able to reliably and robustly preserve causal relationships consistently.