# OpenReview forum: "Causality Preserving Chaotic Transformation and Classification using Neurochaos Learning"
_NeurIPS.cc/2022/Conference — NeurIPS 2022 Accept_

### Official Review · Reviewer_Uusg · 2022-06-15

**Rating:** 4
**Confidence:** 4
**Soundness:** 2 fair
**Presentation:** 2 fair
**Contribution:** 2 fair

**Summary:**

In this paper, the authors utilized a recently proposed Neurochaos Learning method for the classification of cause-effect. To verify the brain-inspired learning method, multiple experiments have been conducted on synthetic and real-world datasets. Compared to a five-layer Neural Network, the utilized NL method achieves superior performance.

**Questions:**

It would be great if the author could address the issues raised in the weakness section.

**Ethics Review Area:**

["I don’t know"]

**Limitations:**

Yes, the authors fairly discussed the limitations of the method. The potential negative impact may not be applicable to this study.

**Strengths And Weaknesses:**

Strengths:
1. The idea of applying a brain-inspired learning method is interesting and has not been extensively explored yet, this could potentially bring novel insights complementary to the mainstream data-driven model design.

2. The experimental results seem plausible and reflect the feasibility of the NL method compared to a baseline neural network.


Weakness:

1 Causality: I think the main drawback of this manuscript is the discussion of causality. In line 25, the authors claim that causality has been mathematically defined by Wiener et.al.. it would be nice to explicitly give the definition here, as reviewers may not familiar with this definition. Importantly, the nuance of causality definition varies from literature [1]  to literature [2]. Without presenting the exact definition of causality quoted in this paper and discussing related definitions, it makes the readers hard to understand the main idea. In terms of 'classification of cause-effect', I am not sure if this terminology makes sense or not. What does it mean by classifying cause-effect (later causality detection is brought in line 38)? I believe the authors should discuss its connection to causal variable identification. This also relates to the fact the study is conducted on observational data.

[1] Peters J, Janzing D, Schölkopf B. Elements of causal inference: foundations and learning algorithms[M]. The MIT Press, 2017.
[2] Hernán M A, Robins J M. Causal inference. 2010.

2 Unclear model design: The model architecture and learning details are fragmented or missing. The authors could either provide a plot of model illustration, pseudo-code table, or code repository. Considering that Neurochaos Learning is not a well-known method, it is important to demonstrate integrated details to facilitate reproductivity.

3 Experimental design: The experiments regarding Coupled autoregressive (AR) processes and Coupled 1D chaotic maps etc. don't seem to be well-motivated. Could the authors reason why particularly using such setting to investigate cause-effect of time series. Lastly, the comparison to a five-layer neural network seems to be less convincing, given the rapid developments of deep learning architectures.

---

> ### Author Response · Authors · 2022-08-02
> **Addressing the weakness comments provided by Reviewer Uusg by conducting new experiments suggested by reviewer**
>
> **Addressing Weakness 1**: We have now included the definition of causality for time-series as has been given by Wiener. This has been included on page 1,  of the revised manuscript and reads as follows:
>
> * According to Wiener, “a time series $X$ causes a time series $Y$, if the past values of $X$ contain information that help predict Y above and beyond the information contained in the past values of $Y$ alone”.
> * Further, in order to clarify the considered task of cause-effect classification, some changes have been made in the Introduction section. We have stated explicitly that we "focus on causality detection from time-series data (without assumption of any causal model)". Also, we clarify the task of "cause-effect" classification and have included the following sentence:
> “To elaborate, the problem of bivariate causality detection is considered, where the algorithm is trained to classify the "causal" or "driver" time-series variable as the cause and the time-series variable that is 'affected' or 'driven' by the former as the effect."
> * Also, from Section 3.1 and Section 3.2, the nature of our cause and effect variables are clear where we give the equations of simulated ‘cause’ and ‘effect’ time series on which the algorithms are to be tested.
> * We do not use the term ‘causal variable identification’ as this term is not used much in the domain of time series causality, but when we have random variables which may not be observed as a function of time.
>
> **Addressing Weakness 2**:  Neurochaos Learning is a recently proposed and well motivated and demonstrated machine learning algorithm. The efficacy of the algorithm has been previously tested on several synthetic and real-world datasets yielding very impressive performance. The earlier work of Neurochaos published renowned journals such as Chaos AIP, Neural Networks etc. We are citing the same for reference here:
> * Balakrishnan, H. N., Kathpalia, A., Saha, S., & Nagaraj, N. (2019). ChaosNet: A chaos based artificial neural network architecture for classification. Chaos: An Interdisciplinary Journal of Nonlinear Science, 29(11), 113125.
> * Harikrishnan, N. B., and Nithin Nagaraj. "When noise meets chaos: Stochastic resonance in neurochaos learning." Neural Networks 143 (2021): 425-435.
> * Laleh, T., Faramarzi, M., Rish, I., & Chandar, S. (2020). Chaotic continual learning, ICML 2020 Workshop LifelongML.
> * The above references are appropriately cited in our manuscript (Ref 22 and 25 in the manuscript). Owing to the space constraints we have not provided the architecture diagram and working of NL (This would be a repetition of Reference 1 and 2 provided above, we have appropriately cited both the references in our manuscript). None of the previous work in NL has explored cause-effect preservation property. We, for the first time, show the causality informed classification property of NL. In order to retain anonymity, we have not provided the GitHub link to the code repository. We will make the GitHub link publicly available in the camera ready submission.

---

> > ### Author Response · Authors · 2022-08-02
> > **Addressing Weakness 3**
> >
> > **Addressing Weakness 3**: We understand the reviewers' concern. AR processes are popular models that simulate the dynamics of timeseries from economics and nature. In the same way, chaotic processes are used to model the dynamics of neurons in the brain, processes in climatology and hydrology. These processes are most often used to study the performance of timeseries causality estimation methods. Hence, we use AR process and chaotic systems to study the cause-effect classification and preservation property of NL. We have also shown an application of NL in preserving the cause-effect relationship of real world prey-predator data.
> > * As per the suggestion of the reviewer, we have used 1D CNN and LSTM to classify the timeseries generated from a coupled chaotic master-slave tent map (Refer section 3.2.1 of the manuscript) and AR process. We compare the performance of NL ChaosNet architecture with five layer Deep Neural Network (DL), 1D Convolutional Neural Network, and Long Short Term Memory.
> > * From the experiments we infer that **NL outperforms DL, CNN, LSTM architectures**. [The performance evaluation figure for classification of cause-effect timeseries generated from coupled chaotic master-slave system is available in this link.](https://drive.google.com/file/d/1LE7GuzSMx5l3O_dwGAremsNHyswzdc40/view?usp=sharing). In the case of CNN, for coupling coefficient = 0, an F1-score = 1.0 is yielded which is actually incorrect. A coupling coefficient = 0 means there is no causal influence from master to slave and hence the classifier should in fact yield a performance score of 50% as the two time series are causally independent from each other (ChaosNet, DL and LSTM are on target here). This implies that CNN is not truly making causally informed classification but rather ‘memorising’ the data. The performance of LSTM was poor compared to all other architectures. However, in the future work, we wish to do extensive hyperparameter tuning for LSTM.
> > * For coupled chaotic master slave system: The preservation of causality is predominantly seen first in NL (a), second in LSTM (d). In the case of DL (b), only for coupling coefficient = 0.2, and 0.3, the cause-effect relationship is preserved. In the case of LSTM, cause-effect relationship is preserved upto a coupling coefficient = 0.3.
> > [Figure: CCC values computed for various coupling coefficients for features extracted using NL, DL, CNN, and LSTM for coupled chaotic master-slave system.](https://drive.google.com/file/d/1heOzb0Xb1aTr_e4HpWCrqMSRjImD2cK1/view?usp=sharing)

---

### Official Review · Reviewer_UvGj · 2022-07-09

**Rating:** 4
**Confidence:** 4
**Soundness:** 2 fair
**Presentation:** 2 fair
**Contribution:** 1 poor

**Summary:**

This paper discusses Neurochaos learning and evaluates how it performs on a classification task where it is asked to identify the cause and the effect.

In brief, Neurochaos learning is this: a 1D skew tent map is a chaotic process. Given an input $x$, we run the process until it comes within $\epsilon$ of $x$. Various statistics for the process are then computed and these are used as the feature representation of $x$. Consider a multi-class classification task. The training data is used to create average-representations, one per class. Classification on the test data is done by computing the cosine similarity with the per-class representations.

The authors consider the task of classifying whether a time series is a cause or effect of another time series. In this task, the authors show that NL representations are found to preserve causality -- meaning, for example, that in the NL feature space, Granger causality increases with more coupled systems, as it should.

Throughout the paper, NL and Deep Learning (DL) are compared. In general, the authors find that for this task, NL and DL achieve similar accuracy, but NL is argued to be more desirable, because of the causality-preserving properties discussed above.

**Questions:**

- Table 1: Why aren't class-0 and class-1 the same number? Aren't the causes and effects generated in pairs?
- line 89: Is the discrimination threshold $b$ the same as the skewness parameter $b$ in equation (1)?
- Section 3.1: What is the train/test split for the AR processes? Or is Table 1 applicable for sections 3.1 and 3.2?
- Section 3: Can you explain briefly what the classification task is? I have two guesses: (1) the classifier is given 2000 steps of a time series and must decide if the time series is the cause or effect. (2) the classifier is given 1000 steps of one time-series and 1000 time-steps of another time-series. The steps are concatenated into a 2000 dimensional vector. The classifier must decide if the first 1000 steps are the cause or the effect. Are either of these correct?
- Section 4.1: How is the data fed as input to ChaosNet? The input is a 2000 dimensional vector. Is each element of the vector fed as stimulus to a single GLS neuron? Or is there a k-dimensional generalization of the 1D neuron presented in section 2?
- line 173: Why were these hyper-parameters chosen?
- Figure 1b: What is the Granger causality computed between? I have two guesses: (1) It is computed between and input time series and the vector of neuron values in the fourth layer (2) It is computed between the average neuron value between the cause and effect time series, after being pushed through the DNN
- Figure 1: What is the difference between "M causes S" and "S causes M"? Is the training data swapped? It's not mentioned in line 173, and unless I'm missing something, it's the first time the terms have been used. My best guess is that "M causes S" is the GC computed between M and S in NL feature-space, where M is the cause and S is the effect. Is this correct?

## minor things
- Line 97: Delete "For"

**Limitations:**

Limitations are adequately discussed.

**Strengths And Weaknesses:**

## Strengths
- Reading about Neurochaos Learning was interesting
- The authors are honest about the limitations of Neurochaos learning and do not omit plots or experiments where DL is shown to perform better. I think this is commendable.

## Weaknessess
- In the introduction, Deep Learning is criticized for having poor interpretability and trustworthiness. But the paper does not present a convincing case that Neurochaos Learning is much better on those fronts. In particular, section 2 and section 4.3 describe the basics of NL prediction and parameter-tuning, but not much interpretation of these are given. It's hard to see how NL can be said to /learn/. From the description given in section 2, the whole process seems to have more in common with non-parametric clustering methods.
- For comparison with Deep Learning, a 5 layer dense feed forward net is used. But given that the task is time-series classification, I feel a more fair comparison would be an RNN, Transformer, or even 1D CNN
- I don't think the results show conclusively that NL captures causality better than DL. Figure 2a shows that Deep learning outperforms NL in as many cases as it underperforms
- The paper claims that the NL representations exhibit the same causal relationships that are present in the input data, but the results presented (figure 2c, figure 5a) don't seem to conclusively show that. Additionally, no baselines are given for a transformation that does /not/ preserve causality, so the quality of the results are hard to judge in figure 2c.
-  My main concern is with significance. The task domain seems extremely tailored to the type of thing NL would be good at. I would be more convinced of the usefulness of NL for causal tasks if it were clear how these methods could be applied to more complex problems. As it is, it's hard to see how this approach would be able to handle, e.g., language modeling.

---

> ### Author Response · Authors · 2022-08-02
> **Overall Response**
>
>  * We thank the reviewer for succinctly summarising the workings of Neurochaos Learning (NL) which is very different from traditional ANN/ML/DL algorithms.  We would like to mention that NL was originally designed for classification tasks (without any thought on causality as we gather from reading the paper).  In this work, it is discovered for the first time that NL also helps in preserving causality in the following scenarios - a) simulated time series from coupled AR processes, b) simulated time series from coupled chaotic tent maps and logistic maps, and c) a real-world predator-prey dataset. We have employed the well known Granger Causality (GC) for determining causal strengths for coupled AR processes and Compression-Complexity Causality (CCC) for coupled chaotic maps and real world time series. Modelling real-world systems using coupled AR processes and/or coupled chaotic systems is common in the domains of Neuroscience, Financial time series, Hydrology, Climate research etc.  Thus, the contribution of our paper is significant considering that we have demonstrated this unique property of NL i.e. preservation of causality as determined by GC and CCC in a number of simulated and real world data sets. Both GC and CCC are used widely in determining causality in the literature pertaining to causality testing. All previous works on NL have not talked about causality and hence our work is the first of its kind and offers scope for a lot of new future research in these directions.
> * We have also done new work after going over the reviewer comments - we have now compared the performance of NL with 1D CNN and LSTM and the relevant graphs are given the response document. We will be submitting these in the supplementary material of the revised manuscript and doing appropriate changes in the main body of the revised manuscript to reflect this new work. The earlier conclusions on NL does not change with this new additional work, but rather only reinforces the power of NL over DL, CNN and LSTM when it comes to causality.

---

> > ### Author Response · Authors · 2022-08-02
> > **Addressing the Weakness comments provided by reviewers**
> >
> > **Addressing Weakness Point No. 1**: Neurochaos Learning (NL) is a supervised brain inspired classification algorithm unlike unsupervised learning clustering algorithms. NL has two architectures ChaosNet and CFX (features extracted from input layer of NL) + ML which have been discussed in Ref. 22 and 25 of the main manuscript. Some aspects of the interpretability of NL has been covered  in Ref 22. This work is also an exploration of the causality related interpretability of NL.
> >
> > **Addressing Weakness Point No. 2**: As per the suggestion of the reviewer, we have used 1D CNN and LSTM to classify the timeseries generated from a coupled chaotic master-slave tent map (Refer section 3.2.1 of the manuscript) and AR process. We compare the performance of NL ChaosNet architecture with five layer Deep Neural Network (DL), 1D Convolutional Neural Network, and Long Short Term Memory.
> > * From the experiments we infer that **NL outperforms DL, CNN, LSTM architectures**. [The performance evaluation figure for classification of cause-effect timeseries generated from coupled chaotic master-slave system is available in this link.](https://drive.google.com/file/d/1LE7GuzSMx5l3O_dwGAremsNHyswzdc40/view?usp=sharing). In the case of CNN, for coupling coefficient = 0, an F1-score = 1.0 is yielded which is actually incorrect. A coupling coefficient = 0 means there is no causal influence from master to slave and hence the classifier should in fact yield a performance score of 50% as the two time series are causally independent from each other (ChaosNet, DL and LSTM are on target here). This implies that CNN is not truly making causally informed classification but rather ‘memorising’ the data. The performance of LSTM was poor compared to all other architectures. However, in the future work, we wish to do extensive hyperparameter tuning for LSTM.
> > * For coupled chaotic master slave system: The preservation of causality is predominantly seen first in NL (a), second in LSTM (d). In the case of DL (b), only for coupling coefficient = 0.2, and 0.3, the cause-effect relationship is preserved. In the case of LSTM, cause-effect relationship is preserved upto a coupling coefficient = 0.3.
> > [Figure: CCC values computed for various coupling coefficients for features extracted using NL, DL, CNN, and LSTM for coupled chaotic master-slave system.](https://drive.google.com/file/d/1heOzb0Xb1aTr_e4HpWCrqMSRjImD2cK1/view?usp=sharing)
> >
> > **Addressing Weakness Point No 3**: We regret the confusion caused in Figure 2.  The following points will clarify:
> >
> > 1.  Figure 2a shows that NL outperforms DL in terms of F1-score for varying coupling coefficients.
> > 2.  Figure 2b and 2c correspond to causality estimated between Master (M) and Slave (S) time series of chaotic coupled skew tent maps in the raw domain and in the NL feature space (firing time).
> > 3.  We have now shown the cause-effect preservation property of NL, DL, 1D CNN, and LSTM. These will be appropriately reflected in the revised manuscript/ supplementary material.
> > 4.  On the other hand, NL is designed for classification purposes but it turns out that causality is preserved between the M and S time series even though this is not taken into consideration in the design of NL. It is purely a feature of NL which is highly desirable in causal machine learning applications.

---

> > > ### Author Response · Authors · 2022-08-02
> > > **Addressing Weakness Point No 5**
> > >
> > > **Addressing Weakness Point No 5**: This is a valid point by the reviewer. However, let us attempt to convince why our paper merits a serious consideration with the following salient points: NL is a very new and recently proposed method in the literature with very promising performance - comparable to state-of-the-art methods in machine learning literature. Please find the below references:
> > > 1. Balakrishnan, H. N., Kathpalia, A., Saha, S., & Nagaraj, N. (2019). ChaosNet: A chaos based artificial neural network architecture for classification. Chaos: An Interdisciplinary Journal of Nonlinear Science, 29(11), 113125.
> > > 2. Sethi, D., & Nagaraj, N. (2022). Neurochaos Feature Transformation and Classification for Imbalanced Learning. arXiv preprint arXiv:2205.06742.
> > >
> > > * Unlike Deep Learning and other traditional machine learning algorithms which boasts of 10,000s of papers, NL has hardly been explored. Our paper, for the very first time, reports a very important feature of NL which has not been reported so far.
> > > * The task domain we have chosen is not tailored for NL to be good at but rather the ones chosen are very relevant and appropriate in a large number of real world applications. Both coupled AR processes and coupled chaotic systems are widely used in modeling real-world systems in neuroscience, financial time series analysis, hydrology, climate research etc. (just to mention a few). Hence, the simulations we have performed are very relevant and in no way designed to favor NL. On the contrary, it is quite a challenging and appropriate testing of both NL and DL that we subject them to by way of these models generating simulated time series.
> > > * Lastly, it must not be forgotten that we have subjected NL and DL to real-world dataset for causality estimation - the predator-prey system (see section 4.7 and results in Table 2).  It is to be noted that NL is able to preserve causal relationships on this real dataset whereas DL fails.
> > > * We have additionally carried out new experiments using 1D CNN and LSTM and find they too have similar issues as DL.  The main inferences of the paper do not change with this additional new work but rather strengthen the case of NL for causal machine learning applications.
> > > * We would very much engage in subjecting NL to further rigorous testing on real world datasets from various domains including language modelling. Recently, there has been one work in testing NL on imbalanced data on a number of real-world datasets with very promising results - including combination of hybrid NL-ML architectures. Please see for further details:
> > > 1. Sethi, D., Nagaraj, N., Harikrishnan, NB., 2022. Neurochaos Feature Transformation and Classification for Imbalanced Learning. arXiv preprint arXiv:2205.06742.
> > > * In the light of the above points we have made, we would like to make a case for a serious consideration of our work so that NL gets noticed in the larger Machine Learning community so that further research work can happen.

---

> > > > ### Author Response · Authors · 2022-08-02
> > > > **Point by point response to the reviewers questions**
> > > >
> > > > **Addressing Question No 1**: Yes, the cause and effect timeseries are generated in pairs. In order to avoid bias, we randomly split the data into training and testing. Because of the random split, the number of data instances in class-0 and class-1 are different.
> > > >
> > > > **Addressing Question No. 2**: Yes, the discrimination threshold and skewness parameter are the same.
> > > >
> > > > **Addressing Question No. 3**: The train-test split for the AR processes is the same as provided in Table 1.
> > > >
> > > > **Addressing Question No. 4**: Classification Task: Given a timeseries, classify whether it is a cause (class-0) or effect (class-1). The length of the timeseries is 2000. Point (1) provided by the reviewer is correct.
> > > >
> > > > **Addressing Question No. 5**: The input layer of NL contains 2000 GLS neurons (number of GLS neurons in the input layer is equal to number of input features in the data). Each element of the 2000 length timeseries is the stimulus to the corresponding GLS neurons (i.e., the k-th element of the timeseries is the stimulus corresponding to the k-th GLS neuron).
> > > >
> > > > **Addressing Question No. 6**: As mentioned in section 4.1, the hyperparameters are obtained by five fold crossvalidation using the training data.
> > > >
> > > > **Addressing Question No. 7**: In order to evaluate the cause-effect preservation property of NL and DL for AR processes, we have carried out the following experiments:
> > > > 1. NL: For a single cause-effect pair data, we extract the firingtime feature corresponding to the cause timeseries and effect timeseries. The Granger Causality is found between the firingtime of cause timeseries and effect timeseries. The GC values are obtained from 50 random trials (50 random cause-effect pairs)
> > > > 2. DL:  For a single cause-effect pair data, we extract the DL features from the second last layer of DL (the layer just before the output layer) corresponding to the cause timeseries and effect timeseries. The Granger Causality is found between the DL features of cause timeseries and effect timeseries. The GC values are obtained from 50 random trials (50 random cause-effect pairs). Please note, we don't take the average value of the DL feature but retain the full vectors for computing the causal relationships.
> > > >
> > > > **Addressing Question No. 8**: We apologize for this confusion.  As it is known, causation is very different from correlation. For one thing, causation is not necessarily symmetric.  Given two time series M(t) and S(t),  the causation from M(t) to S(t) is not necessarily the same as that from S(t) to M(t). In fact, in the simulations we have set up, the Master time series M(t) is always the cause and the Slave time series S(t) is always the effect - owing to unidirectional coupling from the master system to the slave system - for both coupled AR processes and coupled chaotic systems (skew tent maps and logistic maps).  Figure 1a indicates the causation in the NL feature space (firing time) between M(t) and S(t) (in both directions) and Figure 1b indicates the causation estimated in the DL feature space.  As it can be seen, NL feature space preserves the causal structure whereas there is a breakdown of causal structure in DL feature space.
> > > >
> > > > All these points will be addressed in the revised manuscript for better readability.

---

> > > ### Author Response · Authors · 2022-08-02
> > > **Addressing Weakness Point 4**
> > >
> > > **Adressing Weakness Point No. 4**: We have now done more experiments to show that NL representations do a better job at preserving the causal relations present in input data. This is shown in comparison to other methods by checking the efficacy of features extracted from NL, DL as well as CNN and LSTM to preserve the cause-effect relationship as measured by CCC in case of coupled chaotic tent maps and GC in case of coupled AR processes. Since for the case considered in Figure 2, now the causality preserving property of DL as well as other relevant methods are tested, baseline results from other transformations become available, to which NL can be compared and are shown in the figure below.
> > >
> > > [Figure: CCC values computed for various coupling coefficients for features extracted using NL, DL, CNN, and LSTM for coupled chaotic master-slave system.](https://drive.google.com/file/d/1heOzb0Xb1aTr_e4HpWCrqMSRjImD2cK1/view?usp=sharing)
> > >
> > > From the above link to the graph, clearly NL firing time feature preserves the cause-effect relationship much better than other methods in the range of coupling coefficients 0.1 to 0.5, where the master system causes the slave and the slave and the master are not yet synchronized.
> > >
> > > We also show that NL features preserve causality for coupled AR processes much better compared to other methods.
> > > [Figure: GC values computed for features extracted using NL, DL, CNN, and LSTM for AR process.](https://drive.google.com/file/d/1lRKJcMObyw0syG3Ox8DKOB96olAMrvg2/view?usp=sharing)

---

> > > ### Comment · Reviewer_eVRu · 2022-08-07
> > > **Authors' response**
> > >
> > > I was quite clear and confident in y assessment of the paper. I don't think additional experiments are necessary on other architectures to prove causality preservation but strengthened the paper nonetheless. I will stand by initial rating. This is an outstanding paper.

---

### Official Review · Reviewer_eVRu · 2022-07-11

**Rating:** 8
**Confidence:** 5
**Soundness:** 3 good
**Presentation:** 3 good
**Contribution:** 3 good

**Summary:**

The authors have shown that the brain inspired learning algorithm- Neurochaos learning is capable of performing the classification of cause-effect in four different coupled systems namely, coupled autoregressive processes, coupled 1D chaotic skew tent maps, and coupled 1D chaotic logistic maps and predator-prey.  They generated simulated datasets and performed experiments using the Master-slave configuration on three simulated and one real dataset. In addition, the authors showed that Granger causality is preserved under chaotic transformation for the coupled systems. The classification performance of the coupled configuration is compared with 5-layered DL architecture.

**Questions:**

1. Authors have only used F1-score for the performance comparison. I would like to understand from the authors why are the other metrics were not computed like class-wise accuracy or precision.
2. As the training data derived from the different coupled systems are balanced (2000), the authors have experimented on a balanced data set. However it will be important to see how the model performs on unbalanced dataset.
3. Adaptive learning rate – how will the model perform is  a setting where the continuity of tke loss landscape is leveraged instead of treating the Learning rate as a tunable parameter.
4. Authors have experimented on data points of predator (Didinium  nasutum) and prey (Paramecium aurelia) populations,  I suggest experimenting more on datasets like Predator prey:  Lynx and snowshoe hare, Wolves vs rabbits.
5.	The predator prey systems are natural test beds for neuro chaos learning because they are governed by coupled dynamical systems. In this context can the authors comment on the stability of the learning process?


**Ethics Review Area:**

["I don’t know"]

**Limitations:**

NL is not a causal ML algorithm. If the input data has causal structure, NL preserves that structure.  If the problem is not framed as classification of cause effect , then NL requires causality testing methods like GC, CCC etc to identify which is cause and which is effect.
The following are the suggestions:
1. Cause-Effect preservation is an important property for the interpretability of the algorithm. This can be used to understand why the algorithm works. Interpretability of Neurochaos features + ML algorithms can be further investigated using this property.
2. Application of NL in the classification of breast cancer curated dataset with heavy class imbalance may be investigated. Particularly I would like to see a discussion on how NL can be used to detect anomalies where the minority class instances are possibly anomalous, like in the breast cancer data, for specific survival year cut offs.

**Strengths And Weaknesses:**

1. Authors have indicated a classification method of cause and effect for any time series data using the NL algorithm. The method shows a fair performance on the simulated and real datasets when their F1-scores were compared. Clearly, F1-scores for ChaosNet were better than DL.
2. The manuscript also demonstrates the preservation of Granger Causality in the ChaosFEX feature space.
3. Usage of “Transfer learning” like approach by testing the algorithm on a dataset completely different from the one it was trained on.

The aim in the paper is not to build a causal ML algorithm, but rather to explore if the existing NL architecture which does classification can also distinguish between cause and effect data. The study answers this in the affirmative and slightly outperforms DNN in this regard. When we ask an important question regarding features such as whether features extracted from the learning architecture (in the cause-effect classification task) in both DNN and NL preserve causality as measured by Granger Causality (GC) and Compression-Complexity Causality (CCC). This is important because it determines whether NL is doing a causality informed classification or not. For this the authors used GC and CCC to check for causal relationship between coupled AR processes and master-slave chaotic skew tent-map system respectively (Note: In the cause-effect classification task they are not using GC and CCC, but the plain NL architecture which was used for classification). In the experiments, they show unambiguously the failure of GC for DNN features (features extracted from the layer just before the output layer), owing to the failure of cause-effect preservation. Whereas in the case of ChaosNet, using GC and CCC, the cause-effect relationship in the input data was faithfully preserved. This is a verification of causality informed classification - i.e., the output from the input layer of NL preserves cause-effect relationship. This is a very important and novel finding for developing causal NL
algorithms in the future (and not been shown in any of the previously published work on NL).

---

> ### Author Response · Authors · 2022-08-02
> **Comments on new experiments conducted**
>
> We have done new work after going over the reviewer comments - we have now compared the performance of NL with 1D CNN and LSTM and the relevant graphs are given the response document. We will be submitting these in the supplementary material of the revised manuscript and doing appropriate changes in the main body of the revised manuscript to reflect this new work. The earlier conclusions on NL does not change with this new additional work, but rather only reinforces the power of NL over DL, CNN and LSTM when it comes to causality.

---

> > ### Author Response · Authors · 2022-08-02
> > **Point by point response to the questions raised by the reviewer**
> >
> > **Addressing Question 1**: Given that our datasets are nearly perfectly balanced, F1 score is a better measure as it is the harmonic mean of precision and recall and thus is a better performance measure for the classifiers under consideration.
> >
> > **Addressing Question 2**: Point noted. We would be performing these experiments going forward. Thank you for this suggestion.
> >
> > **Addressing Question 3**: Thank you for this suggestion.  This requires a new line of work and thus outside the scope of the present paper. Definitely this is something we will consider going forward in a new line of research.
> >
> > **Addressing Question 4**: Thank you for this suggestion. Yes, we will consider this in future work. We would request the reviewer with a link/pointer/reference to the above datasets.
> >
> > **Addressing Question 5**: Thank you for the comment. The stability of the learning process can be looked at from two points of view (1) Stability of the learning algorithm, (2) Stability of the prey-predator dynamical system. In this case, we can only look at the stability of the learning algorithm (NL) because we don’t know the governing equations of the prey-predator data.
> > One way to check the stability is to add noise to the learned parameters of NL i.e., mean representation vectors. We have to do this experiment to provide a conclusive answer. This is one future direction of NL.

---

> > > ### Author Response · Authors · 2022-08-02
> > > **Addressing Limitation**
> > >
> > > Thank you for this suggestion. Yes, this is definitely worth pursuing in a new line of future research which we will consider. We have also done new work after going over the reviewer comments - we have now compared the performance of NL with 1D CNN and LSTM and the relevant graphs are given the response document. We will be submitting these in the supplementary material of the revised manuscript and doing appropriate changes in the main body of the revised manuscript to reflect this new work. The earlier conclusions on NL does not change with this new additional work, but rather only reinforces the power of NL over DL, CNN and LSTM when it comes to causality.

---

### Official Review · Reviewer_JiUB · 2022-07-22

**Rating:** 4
**Confidence:** 3
**Soundness:** 3 good
**Presentation:** 2 fair
**Contribution:** 3 good

**Summary:**

This paper use Neurochaos Learning architecture for the classification task of cause-effect for data generated from coupled chaotic maps. Results show that NL preserves causality under a chaotic transformation and can successfully classify cause and effect time-series. One more exciting finding is that even a general NL architecture is capable of some essential causal learning. Hence, promising development of more sophisticated causal ML algorithms is required for different tasks.


**Questions:**

1. Lack of novelty: Neurochaos Learning is not first proposed in this work, which may hurt the contribution of this paper.
2. For O3 in #53, transfer learning is not very important in this setting, and a more detailed analysis of the proposed method itself may be helpful. Or the reviewer may want to see why transfer learning needs to be addressed.

**Ethics Review Area:**

["I don’t know"]

**Limitations:**

The authors claimed well in the limitations parts.

**Strengths And Weaknesses:**

Strengths:
1. This paper introduces the Neurochaos Learning architecture (NL) into the causality domain, which can be seen with more applications.
2. The proposed method consistently outperforms a five-layer Deep Neural Network architecture, which is convincing.

Weakness:
1. Lack of novelty: Neurochaos Learning is not first proposed in this work, which may hurt the contribution of this paper.
2. For O3 in #53, transfer learning is not very important in this setting, and a more detailed analysis of the proposed method itself may be helpful.

---

> ### Author Response · Authors · 2022-08-02
> **Comments on new experiments conducted**
>
> We have done new work after going over the reviewer comments - we have now compared the performance of NL with 1D CNN and LSTM and the relevant graphs are given the response document. We will be submitting these in the supplementary material of the revised manuscript and doing appropriate changes in the main body of the revised manuscript to reflect this new work. The earlier conclusions on NL does not change with this new additional work, but rather only reinforces the power of NL over DL, CNN and LSTM when it comes to causality.

---

> > ### Author Response · Authors · 2022-08-02
> > **Point by point response to the questions raised by the reviewer**
> >
> > **Addressing Question 1**: Neurochaos Learning is a recently proposed supervised brain inspired learning algorithm. We list the previous published work in NL by various authors:
> > 1. Balakrishnan, H. N., Kathpalia, A., Saha, S., & Nagaraj, N. (2019). ChaosNet: A chaos based artificial neural network architecture for classification. Chaos: An Interdisciplinary Journal of Nonlinear Science, 29(11), 113125.
> > 2. Harikrishnan, N. B., and Nithin Nagaraj. "When noise meets chaos: Stochastic resonance in neurochaos learning." Neural Networks 143 (2021): 425-435.
> > 3. Laleh, T., Faramarzi, M., Rish, I., & Chandar, S. (2020). Chaotic continual learning, ICML 2020 Workshop LifelongML.
> > 4. Chen, H., Zhang, M., Gao, Z., & Zhao, Y. (2021). Deep ChaosNet for Action Recognition in Videos. Complexity, 2021.
> > 5. Harikrishnan, N. B., & Nagaraj, N. (2019, October). A novel chaos theory inspired neuronal architecture. In 2019 Global Conference for Advancement in Technology (GCAT) (pp. 1-6). IEEE.
> > 6. Harikrishnan, N. B., Pranay, S. Y., & Nagaraj, N. (2022). Classification of SARS-CoV-2 viral genome sequences using Neurochaos Learning. Medical & Biological Engineering & Computing, 1-11.
> > 7. Harikrishnan, N. B., & Nagaraj, N. (2020, July). Neurochaos inspired hybrid machine learning architecture for classification. In 2020 International Conference on Signal Processing and Communications (SPCOM) (pp. 1-5). IEEE.
> > * None of the above published papers have explore the ability of NL to do cause-effect classification for time-series data. In our preliminary research, we were able to find that NL is able to classify as well as preserve cause-effect relationship under feature transformation of timeseries data. We demonstrate that causality preservation property is absent in DL, CNN, and LSTM features. This is the novelty that we are highlighting in our manuscript and the motivation behind this study is well discussed in the Introduction section.
> >
> > **Addressing Question 2**: We have now done a more detailed analysis by comparing our method with other relevant methods such as DL, CNN and LSTM. These results will be included in the supplementary material. Please find the below link to the figure.
> > [Figure: CCC values computed for various coupling coefficients for features extracted using NL, DL, CNN, and LSTM for coupled chaotic master-slave system.](https://drive.google.com/file/d/1heOzb0Xb1aTr_e4HpWCrqMSRjImD2cK1/view?usp=sharing)
> >
> > From the above link to the graphs, clearly NL firing time feature preserves the cause-effect relationship much better than other methods in the range of coupling coefficients 0.1-0.5, where the master system causes the slave and the slave and the master are not yet synchronized.
> >
> > We also show that NL features preserve causality for coupled AR processes much better compared to other methods and also NL does a good job at cause-effect classification of time series data, especially for deterministic chaotic processes. The link to the figure is provided below:
> > [Figure: GC values computed for features extracted using NL, DL, CNN, and LSTM for AR process.](https://drive.google.com/file/d/1lRKJcMObyw0syG3Ox8DKOB96olAMrvg2/view?usp=sharing)
> >
> > Transfer learning is an important aspect of causal learning as has been pointed out in the references below. This is because the assumption that training and test data arise from a common distribution often fails in real world scenarios becoming one of the major limitations of conventional ML algorithms.
> > 1. Schölkopf, B. (2022). Causality for machine learning. In Probabilistic and Causal Inference: The Works of Judea Pearl (pp. 765-804).
> > 2. Peters, J., Janzing, D., & Schölkopf, B. (2017). Elements of causal inference: foundations and learning algorithms (p. 288). The MIT Press.
> > In fact, many specialized causal learning algorithms have been developed to cater to the requirement of ‘transfer’ or ‘out-of-distribution’ learning. Some references are below.
> > 3. Lu, C., Wu, Y., Hernández-Lobato, J. M., & Schölkopf, B. (2020). Invariant Causal Representation Learning.
> > 4. Wang, R., Yi, M., Chen, Z., & Zhu, S. (2022). Out-of-distribution Generalization with Causal Invariant Transformations. In Proceedings of the IEEE/CVF Conference on Computer Vision and Pattern Recognition (pp. 375-385).
> >
> > * In order to motivate the requirement of transfer learning, the following sentence has now been included in the Introduction, page 2:
> > *The motivation behind many specialized causal learning algorithms that have been recently proposed is generalized learning as failure in the case of distribution shifts continues to be one of the most important limitations of traditional ML algorithms [26, 27]. Hence, O3 becomes an important objective to be looked at for an algorithm attempting to learn causal representations.*

---

### Meta-Review · Area_Chair_cfUX · 2022-08-30

**Recommendation:** Accept
**Confidence:** Less certain

**Metareview:**

This paper has been thoroughly evaluated by four competent reviewers. One of them voted strongly for accepting it, while the three others bid borderline rejections. The work tackles an important problem, and it is well written-up. The most important aspect of controversy between the reviewers’ opinions is in whether the proposed method can or cannot preserve causality. Based on my understanding, it appears designed for just that.  The authors have provided extensive rebuttals and, in my opinion, have addressed most of the issues brought up in the initial review. In summary, even though this paper should be rejected based on the straight vote of the reviewers, I would like to encourage the program committee to consider accepting it, provided that there is enough room for it in the program.

**Award:**

No

---

### Decision · Program_Chairs · 2022-09-14

Accept